

# Mass changes of the northern Antarctic Peninsula Ice Sheet derived from repeat bi-static SAR acquisitions for the period 2013-2017

Thorsten C. Seehaus[1], Christian Sommer[1], Thomas Dethinne[2,3], Philipp Malz[1]

[1]Institute of Geography, Friedrich-Alexander-Universität Erlangen-Nürnberg, 91058 Erlangen, Germany
[2]Laboratory of Climatology, University of Liège, Liège, Belgium
[3]Centre Spatial de Liège, University of Liège, Liège, Belgium

*Correspondence to*: Thorsten C. Seehaus (thorsten.seehaus@fau.de)

**Abstract.** Some of the highest specific mass change rates in Antarctica are reported for the Antarctic Peninsula. However, the existing estimates for the northern Antarctic Peninsula (<70°S) are either spatially limited or are affected by considerable

uncertainties. The complex topography, frequent cloud cover, limitations in ice thickness information, boundary effects, and uncertain glacial-isostatic adjustment estimates affect the ice sheet mass change estimates using altimetry, gravimetry, or the input-output method. Within this study, the first assessment of the geodetic mass balance throughout the ice sheet of the northern Antarctic Peninsula is carried out employing bi-static SAR data from the TanDEM-X satellite mission. Repeat coverages from austral-winters 2013 and 2017 are employed. An overall coverage of 96.4% of the study area by surface

elevation change measurements and a total mass budget of -24.1±2.8 Gt/a is revealed. The spatial distribution of the surface elevation and mass changes points out, that the former ice shelf tributary glaciers of the Prince-Gustav-Channel, Larsen-A&B, and Wordie ice shelves are the hotpots of ice loss in the study area, and highlights the long-lasting dynamic glacier adjustments after the ice shelf break-up events. The highest mass change rate is revealed for the Airy-Seller-Fleming glacier system of -4.9±0.6 Gt/a and the highest average surface elevation change rate of -2.30±0.03 m/a is observed at Drygalski

Glacier. The comparison of the ice mass budget with anomalies in the climatic mass balance indicates, that for wide parts of the southern section of the study area the mass changes can be partly attributed to changes in the climatic mass balance. However, imbalanced high ice discharge drives the overall ice loss. The previously reported connection between mid-ocean warming along the southern section of the west coast and increased frontal glacier recession does not repeat in the pattern of the observed glacier mass losses, excluding Wordie Bay. The obtained results provide information on ice surface elevation

and mass changes for the entire northern Antarctic Peninsula on unprecedented spatially detailed scales and high precision, and will be beneficial for subsequent analysis and modelling.

## 1 Introduction and study area

The ice sheet of the Antarctic Peninsula (AP) is strongly affected by the changing climate conditions (e.g. IMBIE Team, 2018; Scambos et al., 2014). A pronounced rise in the air temperature along the AP was reported in the 20th century (Turner

et al., 2016). However, since the turn of the millennia a cooling trend was observed (Oliva et al., 2017; Turner et al., 2016).



Recent analysis suggests an end of the intermediate cooling and the return of a temperature increase (Carrasco et al., 2021). The record summer temperatures measured at stations on the northern AP in the last years are in line with this finding.

Large parts of the coastline of the AP are surrounded by ice shelves, buttressing the ice discharge of the tributary glaciers. Between the 1950s and 2010s, about 28.000 km² of the ice shelf area was lost (Cook and Vaughan, 2010). Most notable are
the disintegration of ice shelves, like Larsen-A and Prince Gustav ice shelves in 1995, Larsen-B Ice Shelf in 2002, and the recession of Wordie Ice Shelf since the 1960s (Wendt et al., 2010). As a consequence, the former tributary glaciers reacted by further frontal retreat, increased flow speeds, and ice mass loss due to the loss of the frontal buttressing (e.g. Friedl et al., 2018; Rott et al., 2011; Scambos, 2004; Seehaus et al., 2015; Wuite et al., 2015). Various studies suggest that the atmospheric warming on the AP in the 20th century has triggered these events (Scambos et al., 2003; Turner et al., 2016;
Vaughan et al., 2003). Moreover, higher basal melt rate caused by warming ocean water might have thinned and weakened the ice shelves before their collapses, as predicted for Larsen C Ice Shelf (Hogg and Gudmundsson, 2017; Holland et al., 2015). Another phenomenon affecting the AP ice sheet is the up-welling of warm Circumpolar Deep Water (CDW) along the south-west coast of the AP (Holland et al., 2010), potentially leading to increased subaqueous melt, frontal recession, and ice discharge (Cook et al., 2016; Hogg et al., 2017; Walker and Gardner, 2017; Wouters et al., 2015).

These different processes are the main drivers of the observed increase in ice mass loss on the AP from 7±13 Gt/a in the period 1992-1997 to 33±16 Gt/a in the period 2012-2017 (IMBIE Team, 2018). Even though, the analysis by the Ice Sheet Mass Balance Inter-comparison Exercise (IMBIE) team relies on mass balance estimates from various methods (altimetry, gravimetry, input-output methods), the reported values are affected by considerable uncertainties. The mean mass budget estimates of the different methods have uncertainties of up to 90% and differ by up to 500%. Frequent cloud cover and the
complex topography along the AP, especially in the regions north of 70°S, imply limitations for altimetric measurements (Schröder et al., 2019; Shepherd et al., 2019). The gravimetric glacier mass budget estimates stretch from -39 to -9 Gt/a with uncertainties in the range of 1-24 Gt/a (IMBIE Team, 2018). These limitations can be attributed to the small West to East extent of the AP, mass changes on the surrounding island as well as uncertain regional glacial-isostatic adjustment (GIA) estimates of the Earth crust (Horwath and Dietrich, 2009). Within the overlap period (2002-2010), both input-output method-
based mass balances estimates for the AP, employed by the IMBIE assessment, differ by up to 30 Gt/a, which is comparable to the mean mass budget in the period 2012-2017. The uncertainty of the input-output method is mainly caused by the uncertainty of the modelled climatic mass balance (CMB) and the accuracy of the available ice thickness data, which has certain limitations on the AP (Seehaus et al., 2015), used to compute the ice flux (IMBIE Team, 2018).

Studies on regional and mountain range scales (Abdel Jaber et al., 2019; Malz et al., 2018, Seehaus et al. 2020) as well as on
continental to global scale (Braun et al., 2019; Brun et al., 2017; Dussaillant et al., 2019; Hugonet et al. 2021) highlighted the suitability and accuracy of the geodetic method. There is currently no geodetic mass balance estimation covering large regions of the AP, like the drainage basins defined by Rignot et al. (2011) or Zwally et al. (2012). Scambos et al. (2014) provided the most extended geodetic mass balance computation on the AP, partially covering regions north of 66°S for the primarily period 2003-2008. The authors used SPOT5 and ASTER stereo imagery, in combination with ICESat-1 altimeter





data. Due to the frequent cloud cover on the AP, analyses based on optical satellite data are less suitable due to limited coverage (Dussaillant et al., 2019; Hugonet et al., 2021). Whereas analyses based on interferometric SAR data (e.g. Malz et al., 2018; Seehaus et al., 2019) are not limited by the weather conditions. Since 2011, the bistatic Synthetic Aperture Radar (SAR) satellite mission TanDEM-X (TDX) has been acquiring data along the AP. Several complete coverages of the AP were acquired for the "global DEM" and "change DEM" missions of TanDEM-X. Various studies showed the feasibility of

obtaining geodetic mass balances on the AP on glacier and multi-glacier scales (Rott et al., 2018, 2014; Seehaus et al., 2015, 2016). Consequently, this study aims to carry out the first large-scale geodetic mass balance analysis on the AP based on TDX data.

The study area is limited to the AP Ice Sheet north of 70°S, excluding the surrounding islands (see Figure 1). This spatial extend was selected, since (1) this section of the AP is strongly affected by the disintegration of ice shelves, (2) up-welling

of CDW along the west coast, and (3) limitations of mass budget estimates based on altimetry, gravimetry and input-output method (see above). An area-wide geodetic mass balance assessment based on TanDEM-X data will provide unprecedented spatially detailed and precise analysis, and will be highly spatially complementary to the results based on other approaches for the more southern sections of the AP.

## 2 Data

In order to obtain surface elevation information at the study site, bistatic Synthetic Aperture Radar (SAR) acquisitions from the TanDEM-X mission were used. Several, most partial, TanDEM-X data coverage of the AP exist since 2011. The surface conditions affect the SAR signal penetration depth in snow and ice (Abdullahi et al., 2018; Rott et al., 2021). A seasonal variability of the mean glacier surface height of about 2 m was reported for the north-eastern AP using TDX data (Seehaus et al., 2015, 2016). However, according to Rott et al. (2018) difference in the SAR signal penetration of TDX can be neglected

on the AP when comparing data from winter seasons. This assumption is based on comparing elevation changes from repeated TanDEM-X acquisitions and repeated airborne Lidar measurements from NASA's Operation IceBridge. Consequently, we used two coverages of the AP with TDX data acquisitions from austral-winter 2013 and 2017 for our analysis. A small data gap in the coverage from 2013 (9.6%) was filled by TDX data from austral-winter 2014 (see Figure 2). A reference DEM (refDEM) is needed for the generation of SAR DEMs based on differential interferometry. The recently

published high-resolution DEM of the AP (Dong et al., 2021) based on the global TanDEM-X DEM at 12 m spatial resolution is employed. The authors used information from the Reference Elevation Model of Antarctica (REMA) (Howat et al., 2019) to correct residual systematic elevation errors in the global TanDEM-X DEM, and to obtain enhanced surface height data for the AP. The temporal coverage of the Data used to generate the refDEM is comparable to our TDX coverage in 2013. However, no pixel-specific date information is available for the global TanDEM-X DEM and consequently for the

refDEM, justifying the need to reprocess a surface elevation model for this time step.



Output from the regional climate model MAR ("Modèle Atmosphérique Régional" in French) covering whole Antarctica is used to obtain information on the CMB. MAR is a polar-oriented climate model mostly used to study the Greenland (Delhasse et al., 2020; Fettweis et al., 2021) and Antarctic ice sheets (Amory et al., 2021; Gilbert and Kittel, 2021). Hydrostatic approximation of primitives equations described in (Gallée and Schayes, 1994) are the basis of the atmosphere

dynamics of the model and its radiative transfer scheme is adapted from (Morcrette, 2002). The energy and mass transfer between the atmosphere and soil is handled by the SISVAT module (Soil Ice Snow Vegetation Atmospheric Transfer (De Ridder and Gallée, 1998)). For this study, the version used is MARv3.12, for which improvements have been described in (Lambin et al., 2022). MAR was run over the AP at a 7.5 km spatial resolution and has been set to resolve the first 20 m of the snowpack, divided into 30 layers of varying thickness. The model is forced by the 6-hourly ERA-5 reanalysis (Hersbach

et al., 2020) at the lateral boundaries and over the ocean between March 2006 and May 2022 but the data up to 2008 has been discarded as spin-up. The snowpack is initialized from a previous simulation (Kittel et al., 2021). The parametrization and evaluation of the model over the AP are described in (Dethinne et al., 2023).

The average CMB is computed for the period July 2013 until June 2017 and the absolute and relative difference (dCMB) in respect to whole temporal coverage of the MAR data (2008-2022) is computed to obtain information CMB anomalies during

the study period.

By dividing the CMB anomalies (dCMB) by the total mass change ($\Delta M/\Delta t$), we define the mass balance ratio MBR. It indicates the contribution of CMB changes on the mass change. Positive values indicate that dCMB and $\Delta M/\Delta t$ are aligned, e.g. decreased CMB and total mass loss, whereas negative values point out contrary alignment, e.g. increased CMB and total mass loss. MBR values close to 1 indicate that the total mass changes can be mainly attributed to changes in CMB

Information on individual glacier outlines, rock outcrops, and regional drainage basin definitions are taken from Silva et al. (2020), Rignot et al., (2011), and Zwally et al. (2012).

## 3 Methods

The TDX data was ordered in Coregistered Single Look Complex (CoSSC) format. Consecutive acquisitions from the same date and orbit were concatenated to enhance the subsequent SAR processing and coregistration of the products. The

differential interferometric SAR processing approach was applied to derive DEMs from the TDX data by means of the refDEM. The advantage of this approach is, that only the elevation difference between the refDEM and the TDX data needs to be unwrapped, leading to fewer phase-unwrapping issues in areas with complex topography like the AP. The resulting TDX raw DEMs need to be coregistered to remove residual horizontal and vertical offsets, in order to generate a smooth DEM mosaic for each time step and to facilitate the comparison of the DEMs from different dates. The iterative

coregistration procedure consists of phase-ramp removal operations and 3D-coregistration based on the algorithm of Nuth and Kääb (2011). More details on the SAR processing and the coregistration procedure can be found in Sommer et al. (2022).



In previous studies (e.g. Braun et al., 2019; Malz et al., 2018; Sommer et al., 2022), the applied coregistration procedure was primarily based on the offset estimation between the refDEM and the TDX DEM on stable areas (ice-free land surfaces). At the AP the amount of ice-free areas is very limited. Less than 4% of the surveyed area is not covered by glacier ice according
to the rock masks from Silva et al. (2020). Moreover, many of the ice-free areas are situated at steep slopes, where DEMs are typically less reliable (Toutin, 2002), or SAR layover and shadow limit the availability of elevation data in the individual raw TDX DEMs. Since the refDEM was generated based on TDX acquisitions in austral winters 2013 and 2014, it is assumed that the elevation differences between the refDEM and our TDX DEMs in 2013/14 are minimal in most ice-covered areas. In particular, away from the dynamic glacier tongues and low-lying areas, previous studies reported only minor
elevation change rates (Scambos et al., 2014; Seehaus et al., 2016). Consequently, the lower sections (<300 m a.s.l.) and dynamic glacier sections, manually defined by means of NASA MEaSUREs ice velocity mosaics (Mouginot et al., 2017), were masked out and the remaining ice-covered areas were included in the coregistration of TDX DEMs from 2013. The difference between the coregistered TDX DEMs and refDEM revealed some areas with remaining systematic elevation differences, e.g. caused by phase-unwrapping issues, in areas selected for coregistration. Those areas were manually
inspected and masked out by an iterative process. For the TDX DEMs in 2017, also an iterative procedure of coregistration relative to the refDEM was applied, consisting of DEM differentiating and updating the masks. At later processing steps, potential biases caused by SAR signal penetration difference were observed at high elevated areas (see below), consequently these areas were also excluded in the iterative coregistration process. While doing this iterative masking, systematic elevation offsets were present in the elevation differences between the refDEM and both TDX DEM mosaics in some areas,
mainly around the Larsen-B embayment (see Figure 2c&d). The offsets showed some pattern which can be attributed to mosaicking of individual DEM tiles. The pattern does not fit to the outlines of the individual DEMs used to generate the TDX DEM mosaics in this study. Thus, it is concluded that remaining residual systematic elevation biases in the refDEM caused these offsets. Consequently, the affected areas were masked out during the coregistration process.

The resulting coregistered TDX DEM were mosaicked for both time steps and subtracted to obtain elevation change (dh)
information. Mosaics consisting of pixel-wise date information were also generated to allow for a precise definition of elevation change rates ($\Delta h/\Delta t$). Subsequently, the ice mass change rates ($\Delta M/\Delta t$) were computed using the Antarctic Ice Sheet basin definitions 25g and 26g from Zwally et al. (2012) and I-Ipp from Rignot et al. (2011). Additionally, the basin delineations were cropped in latitudinal subsets in 1° steps to investigate potential spatial variations. The ice mass changes were also computed for individual glacier basins larger than 20 km², using the most recent glacier outlines in the glacier
inventory of Silva et al. (2020), and other local sub-region definitions for comparison with existing studies (Rott et al., 2018; Scambos et al., 2014). Voids in the elevation change field on glacier areas were filled using local hypsometric interpolation for the analysis of the individual glaciers throughout the AP, which is one of the most suitable methods according to (Seehaus et al., 2020b). The global hypsometric interpolation was applied for the analysis on basin and sub-region scales. For all ice volume computations, the rock-outcrop definition from Silva et al. (2020) was applied to mask out ice-free areas
in the different ice sheet basins definitions. The ice volume changes were converted to mass changes using a volume-to-mass





conversion factor of 900 kg/m³. Since the most dominant ice volume changes are found for the various former ice shelf tributaries (see Figure 1), this scenario is a suitable factor for ice mass changes dominated by ice dynamics (Kääb et al., 2012).

According to Rott et al. (2018), differences in the SAR signal penetration can be neglected, when comparing TDX DEMs
from austral winters on the AP. Even though, solely TDX austral winter data is used in this analysis, some elevation change patterns in upper glacier regions (see Figure 1 and 3) seems to be caused by differences in the SAR signal penetration between the acquisitions. These areas are located at elevations above 1800 m a.s.l. covering an area of 12.306 km², corresponding to 16.4% of 25g & 26g drainage basins. In order to correct for these potential offsets, we applied a linear increasing correction of dh for the elevation range from 1800 – 2400 m a.s.l. of up to 2 m, similar to Braun et al. (2019) or
Seehaus et al. (2020a), leading to correction of the volume change by -2.76 km³ and an average elevation change by 0.04 m/a (25g & 26g basins). We are comparing X-band to X-band SAR data, in contrast to Braun et al. (2019), who compared X-band to C-band SAR data and applied a variable correction value of up to 5 m. Thus, a reduced maximum correction value of 2 m was selected based on the findings by Seehaus et al. (2015),

Finally, the ice mass change rate $\Delta M/\Delta t$ is computed by:

$$\frac{\Delta M}{\Delta t} = \rho \left( \int_S \frac{\Delta h}{\Delta t} dS + \frac{V_{pen}}{\Delta t} \right) \quad (1)$$

Where S is the analysed glacier area, Vpen the volume change correction to account for differences in SAR signal penetration at higher elevations (see above), and $\rho$ the volume-to-mass conversion factor. The uncertainty of ice mass changes is computed by:

$$\delta_{\Delta M/\Delta t} = \sqrt{ \left(\frac{\Delta M}{\Delta t}\right)^2 \left( \left(\frac{\delta_{\Delta h/\Delta t}}{\frac{\Delta h}{\Delta t}}\right)^2 + \left(\frac{\delta_S}{S}\right)^2 + \left(\frac{\delta_\rho}{\rho}\right)^2 \right) + \left(\frac{V_{pen}}{\Delta t} * \rho\right)^2 + \left(\frac{V_{int}}{\Delta t} * \rho\right)^2 } \quad (2)$$

Where $\delta\Delta h/\Delta t$ is the uncertainty of the elevation change measurements and $\delta$S is the uncertainty of the glacier outlines. Even though, the ice loss is dominated by ice discharge, we account for uncertainties in the the volume-to-mass conversion factor due to surface processes by applying a $\delta\rho$ of 60 kg/m³, according to Huss (2013).

To account adequately for the SAR signal penetration bias correction in the error budget of the mass changes, a 100% uncertainty of Vpen is assumed. Vint is the uncertainty caused by the interpolation of $\Delta h/\Delta t$ in areas without $\Delta h/\Delta t$
measurements. It is computed by multiplying the glacier area with interpolated $\Delta h/\Delta t$ values by the uncertainty of $\Delta h/\Delta t$ caused by the interpolation (0.09 m/a and 0.14 m/a for local and global hypsometric interpolation, respectively), which is computed according to Seehaus et al. (2020b).

$\delta\Delta h/\Delta t$ was estimated based on slope-weighted elevation differences on ice-free areas and considering spatial auto-correlation according to Rolstad et al. (2009), using a correlation length of 318 m (Sommer et al., 2022). In order to account
for potential local differences in the accuracy of the elevation changes, only ice-free areas within the individual ice sheet drainage basin, including the latitudinal subsets, were considered for the analysis of the different basins. Most individual glacier basins have very limited ice-free areas. Thus, the slope-weighted elevation differences revealed on the study area



wide ice-free surfaces were used for the analysis on glacier scales. However, the individual glacier or basin area was used to account for spatial auto-correlation.

Since the ice sheet basin definitions from Zwally et al. (2012) and Rignot et al. (2011) are fixed standard products for mass balance computations in Antarctica, dS was set to zero when using these basin delineations. However, for the analysis of the individual glaciers dS was estimated using the length of the ice-ocean glacier boundaries time a horizontal uncertainty ± 60 m (reliability rating of 1 according to Ferrigno et al. (2006)).

## 4 Results and Discussion

The revealed surface elevation information covers 96.4% of the glaciated area of the northern AP Ice Sheet (basin definition 25g&26g) and is illustrated in Figure 1. The spatial distribution of the ice surface elevation changes clearly indicates local hot spots of ice mass losses for some former ice shelf tributaries of the Larsen-A and B, and Wordie ice shelves. This spatial pattern fits also to altimeter-based observations like (Schröder et al., 2019; Shepherd et al., 2019). It is also clearly visible on the glacier scale analysis of the ice mass changes as illustrated in Figure 4. Surface lowering rates of up to -8 m/a and more,

as well as glacier-wide mean surface elevation change rates of -0.71±0.01 m/a, -2.30±0.03 m/a, and -1.99±0.05 m/a are found for Airy-Seller-Fleming (ASF), Drygalski, and Hektoria-Green-Evans (HGE) glaciers, respectively, which account for almost 40% of the total mass loss of the study area. The overall highest ΔM/Δt value of -4.9±0.6 Gt/a is found for ASF Glacier, which is the largest glacier (7710 km²) in the study area. The highest average surface lowering rate is observed at Drygalski Glacier. For a few, mainly small, glaciers there are also slight positive mean Δh/Δt values measured. Cook et al.

(2016) proposed for the south-western coast of the AP a correlation between frontal retreat and the warming of mid-ocean water layer, due to up-welling of CDW, since the 1990s. Walker and Gardner (2017) as well as Friedl et al. (2018) attributed the recession and increased ice discharge at Wordie bay to the same phenomena. Our results confirm these propositions at Wordie Bay, where high ice losses are measured. However further north, there is a very heterogeneous change pattern revealed on glacier scales (Figure 4). By averaging the ice sheet changes on 1° latitudinal scales (Figure 5 and Table 1) there

is also no correlation observed, regarding the warming pattern reported by Cook et al. (2016). There is to note that the differences in the observation periods of this study and by Cook et al. (2016) (1945-2009) might explain the discrepancies. On the other hand, most glaciers along the west coast are situated in fjord-like valleys. Thus, the frontal retreat might not have destabilized the ice discharge. In order to test this hypothesis, further studies on the evolution of the ice flow are needed. Even though, for section 66-67°S an average slight mass gain, and for section 68-69°S moderate ice loss is found. The

comparison of ΔM/Δt with the average CMB and dCMB values (Figure 5) on latitudinal scales does not show a correlation pattern. The dCMB indicates, that the average CMB throughout the study period was lower in the southern and higher in the northern section of the study site in comparison to the long-term average CMB (Figure 5e). The MBR values obtained in this study are illustrated in Figure 5b and listed in Table 1. Negative MBR is revealed for the northern part of the study area, indicating that changes in CMB were not the driver of mass losses. However, for wide parts of the Larsen-C tributaries along

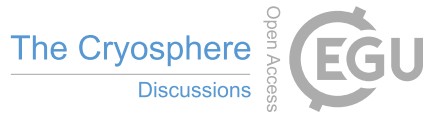

the east coast and the southern section of the west coast positive MBRs are revealed, suggesting that the decrease in CMB contributed to the total mass losses, in particular for the section between 67° and 69°S on the west coast. A negative MBR is revealed for section 66-67°S on the west coast. Here, a slight total mass gain and negative dCMB are found. Thus it can be assumed that reduced ice discharge might have compensated the lower CMB and even led to a mass gain. It is noteworthy that the modelled CMB is subject to considerable uncertainties, which can be assumed in the range of 14-17% on the AP

(Rignot et al., 2019), and that the revealed $\Delta M/\Delta t$ values have certain error margins as well. Thus, further analysis of the ice dynamics is needed to back up the drawn conclusions on correlations between CMB and $\Delta M/\Delta t$, which is beyond the scope of this study.

The Imbalance ratio, defined by Scambos et al. (2014) as the mass change divided by the CMB, serves as an indicator for the ice mass imbalance. It is illustrated in Figure 5c and listed in Table 1. Its spatial pattern generally repeats the $\Delta M/\Delta t$ and

MBR pattern and also clearly indicates the continuous high imbalance of the regions affected by the past ice shelf break-up events.

By splitting the study side into subregions based on the glacier front type (see Figure 6), average $\Delta h/\Delta t$ values of -0.317±0.004 m/a, -0.123±0.004 m/a and -0.171±0.006 m/a are found for former ice shelf tributaries, current ice shelf tributaries, and non-ice shelf tributaries, respectively (Table 1). It indicates that the aftermath of the ice shelf breakup events

forces increased mass losses of former tributaries even throughout multiple decades, accounting to 67% of the study area wide mass loss.

Along the coastline of the Larsen-A and B and Wordie embayment, higher glacier flow speeds of the former ice shelf tributaries are reported by Rott et al. (2018), Seehaus et al. (2018), and Friedl et al. (2018) for the study period of this analysis. The observed ice thickness changes by this study are in accordance with the remaining accelerated ice discharge of

these glaciers, which indicates imbalance conditions. The most pronounced accelerated ice flows as compared to the pre-collapse conditions were reported for Boydell, Sjögren, Drygalski, and Hektoria-Green Glacier. These observations fit well with the high ice thickness change rates in the range of -1.25 to -8.84 m/a found at these glaciers within this study and reported by Rott et al. (2018) for the period 2013-2016. Rott et al. (2018) observed a total mass budget for the analysed glacier area (6358.7 km², Figure 6) of -8.668 ±1.005 Gt/a (1.514 ± 0.176 m/a), which is comparable to our result (observed

area: 11685.6 km²) of -7.3 ±0.6 Gt/a (-0.69±0.01 m/a). The total mass budget agrees well. However, the average elevation change rates differ considerably, which can be explained by (1) slightly different observation periods, (2) different glacier outlines, (3) and most important, Rott et al. (2018) analysed only the lower, dynamic outlet glacier tongues (6358.7 km², as compared to 12723.4 km² of the total area of all glaciers, based on the outlines used by the authors). Based on this comparison it can be concluded that the glacier tongues dominate the mass losses of the former tributaries due to the

disintegration of the Larsen-A and B ice shelves and that the higher elevated plateau regions show now considerable changes, which is in accordance with Scambos et al. (2014) and our assumption for the coregistration process of the raw TDX DEMs. Moreover, the elevation change patterns on Hektoria-Green Glacier revealed in this analysis support the assumption that wide parts of the lower glacier sections are floating. The distinct change from high surface lowering rates to much smaller





rates towards the calving front, indicates that the surface lowering signal by the thinning of the ice is widely compensated by
the buoyancy of the ice. A similar, but less pronounced, pattern is also visible at Dinsmoor-Bombardier-Edgewoth (DBE)
glacier system and was already discovered by Seehaus et al. (2015). The proposed grounding line position based on our
elevation change analysis is also comparable to the one suggested by Rott et al. (2020) for 2016, based on mapping a break
in slope for Hektoria-Green Glacier (see Figure 7). At Crane Glacier Rott et al. (2020) suggested as well as grounding line
position for 2016. However, our elevation change pattern does not allow a reasonable mapping of such a feature (note: the
observed pronounced elevation increase towards the calving front is caused by frontal re-advance). At Wordie Bay (see
Figure 7), episodic glacier retreat and acceleration of Fleming Glacier were reported by Friedl et al. (2018), indicating its
imbalance. Their suggested grounding line position for 2014 fits partly to the elevation change pattern observed at the lower
glacier sections in this study (see Figure 7), similar to Hektoria-Green Glacier.

Scambos et al. (2014) carried out the spatially most extended (<66°S) geodetic analysis on the AP of glacier elevation and
mass changes primarily for the period 2003-2008. Due to the difference in the observation periods and the inclusion of the
surrounding islands in the regional analysis, a direct comparison of their finding with ours is difficult, in particular, due to
dynamic changes of the former ice shelf tributaries along the east coast. However, the general spatial pattern is quite similar.
They reported a total mass loss rate of -24.9 Gt/a, -4.7 Gt/a, -2.3 Gt/a, and -18.0 Gt/a for the nAP <66°S, nAP West, nAP
North, nAP East (excluding the islands: -20.4 Gt/a, -3.9 Gt/a, -1.8 Gt/a, -14.8 Gt/a; see Scambos et al. (2014) and Figure 6
for region definitions), respectively. Whereas, we observed -11.0±1.2 Gt/a, -3.0±0.7 Gt/a, -0.1±0.0 Gt/a, and -7.8±0.6 Gt/a,
respectively. Both analyses obtained similar moderate mass losses for the western sections. For the relatively small nAP
North section the difference might be caused by the very limited coverage of the area by DEM data in the study by Scambos
et al. (2014). Elevated mass losses are found for the east coast by both analyses, which can be attributed to the imbalance of
the former ice shelf tributaries in this section. Reduced mass losses are revealed for the more recent observation period by
this study, which is in accordance with other analyses (e.g. Rott et al., 2018; Seehaus et al., 2018).

On drainage basin scales a comparison with altimetry, gravimetric, and input-output method-based estimates is feasible. Our
revealed results for the different drainage basin definitions are summarized in Table 1. The gravimetric assessment of the
mass budget of Antarctica by Groh and Horwath (2021) (https://data1.geo.tu-dresden.de/ais_gmb/) suggests a mass balance
for the AIS28 basin definition (corresponding to the 25g and 26g basins) of -5.2 ± 48 Gt/a for the period 2013-06-16 to
2017-06-10, whereas we observed -24.1 ± 2.8 Gt/a. Even though both estimates agree within the error budget, there is a
considerable difference in the nominal value. The huge uncertainty of the gravimetric-based estimate clearly indicates the
limitations of this approach for the study region. Moreover, the coarse spatial resolution does not allow to resolve detailed
spatial patterns or even the analysis of the mass budget on glacier scales. The comparison with altimeter measurements is
difficult since most analysis report results only on ice sheet basins scales and for much larger observation periods (e.g.
Schröder et al., 2019; Shepherd et al., 2019). However, a meaningful comparison with Smith et al. (2020) is possible. The
authors reported a mass budget of -10±2 Gt/a and -16 ± 3 Gt/a for the 25g and 26g basin definitions for the period 2003-2019,
respectively. For basin 25g the estimates agree well within the uncertainties with our findings. For basin 26g, they revealed



higher mass loss rates. Their analysis starts in 2003, shortly after the disintegration of Larsen-B Ice Self and only a few years after the break-up of Larsen-A and Prince-Gustav-Channel ice shelves. Consequently, the dynamic ice mass loss of the former ice shelf tributaries located in basin 26g and the subsequent adjustments explain the difference to our estimate. Even though the uncertainties are much lower and the spatial resolution is higher than the altimeter based estimates (e.g. 10 km for Schröder et al. (2019), 5 km for Shepherd et al. (2019)) compared to the gravimetric results, the revealed elevation change maps show a blurry pattern hampering more spatially detailed analyses.

Rignot et al. (2019) provides mass balance estimates based on the input-output method on ice sheet basin scales, but also for a few individual glaciers. Thus a comparison on glacier scales is possible, where the individual glaciers could be clearly identified in the inventory used by them and by this study. Based on the supplementary information provided by Rignot et al. (2019), the mass budget was calculated by subtracting the average ice discharge in the years 2013-2017 from the reported reference SMB. A summary of the comparison with our results, mainly covering glaciers in the Larsen-B embayment, is provided in Table 2. The considerable difference between our estimates of the total mass balance and the results by Rignot et al. (2019) can be partly attributed to differences in the glacier basin definitions. To compensate for the partially strong differences in the glacier basin areas, the specific mass balances were computed. However, for the specific mass balances there are considerable deviations between both estimates as well, even though our estimates agree well with the other geodetic estimates (see above). The limitations in reliable ice thickness estimates towards the grounding lines on the AP (see e.g. Seehaus et al., 2015) and the applied assumptions to overcome these limitations by Rignot et al. (2019) are supposed to strongly bias the mass balance estimations. In particular the assumed balanced conditions for some of the glaciers (Mapple, Pequod, Punchbowl, Starbuck, and Stubb Glacier) by the authors needs to be considered with care. For the SCAR-Inlet tributaries, namely Starbuck and Stubb, and for Pequod Glacier it might be a suitable assumption, however for Mapple and Puchbowl Glacier, this assumption should be revised, considering the specific mass balance estimates (see Table 2). On ice sheet basin scales, a mass budget of -30±28 Gt/a for the I-Ipp basin is revealed by the input-output method, which overlaps with our estimate within the considerable error margins. It can be assumed, that the result is most likely biased by the assumptions made to compensate for the missing good-quality ice thickness information in the study area.

## 5 Conclusions

By using repeated coverages of the northern AP by bi-static SAR data from austral winters in 2013 and 2017, it was possible to obtain a nearly full coverage (96.4%) of ice surface elevation change measurements throughout the study area. The revealed spatial pattern of glacier changes and the overall mass budget of -24.1±2.8 Gt/a agree well with other analyses. The detailed comparison of the revealed glacier changes at the Larsen-A and B, and Wordie embayments with other published data, based on elevation change measurements, highlights the suitability of the applied approach and the quality of the obtained results. However, the comparison with estimates based on the input-output method revealed strong deviations, in



particular on glacier scales. These findings stress the need for improved ice thickness data towards the grounding line along
the AP, which is the dominating error source in ice discharge estimates on the AP.

By including information on climatic mass balance, it could be identified that the observed mass changes can be, at least partly, attributed to climatic mass balance variations, for wide parts of the southern section of the study area. However, most of the revealed mass losses are caused by ice dynamic changes. In particular, the still ongoing increased ice discharge at the former ice shelf tributaries at the Prince-Gustav-Channel, Larsen-A&B, and Wordie ice shelves are the hot-spots of mass
loss, and 67% of the total mass loss throughout the study area can be attributed to these regions.

The previously reported correlation between increased frontal recession and mid-ocean warming along the south-western coast of the study area could not be repeated by the surface elevation or mass change pattern observed in this analysis, excluding Wordie Bay. Probably, the ice flow of the well-confined glacier tongues in the fjord-like valleys did not get destabilized by the frontal retreat. In order to backup this assumption and to further analyse the obtained glacier changes and
its driving factors, a detailed analysis of the evolution of the ice dynamics throughout the study area would be desirable, which is, however, beyond the scope of this analysis.

This study provides the first geodetic assessment of glacier mass balances base on DEM differentiating throughout the northern AP at unprecedented spatially detailed scales and with high precision. The findings allow ice elevation change and mass budget estimates on ice sheet basins as well as individual glacier scales, which will be beneficial for glaciological
modelling, like enhanced ice thickness reconstructions, and continental and global estimates of ice mass changes and sea level rise computations.

**Author Contributions:** TS initiated, designed and led the study, processed and analysed the data and wrote the manuscript. TS, PM and CS developed jointly the analysis routines for elevation change and mass balance computations. TD contributed
the MAR data and CMB analysis. All authors revised the manuscript.

**Competing interests:** The authors declare no conflict of interest. The founding sponsors had no role in the design of the study; in the collection, analysis, or interpretation of data; in the writing of the manuscript, and in the decision to publish the results.

**Acknowledgements:** This work was financially supported by the ESA Living Planet Fellowship MIT-AP granted to TS and
the priority programme "Regional Sea Level Change and Society" by the grant DFG BR2105/14-2. The authors would like to thank the German Aerospace Center for providing TanDEM-X data free of charge under AO XTI_GLAC0264. A. Silva kindly provided the glacier inventory of the AP from Silva et al. (2020). P. Friedl kindly provided the groundling line data from Friedl et al. (2018). The TanDEM-X DEM of the AP used as reference was kindly provided by D. Floricioiu. ERA-5 reanalysis data (Hersbach et al., 2020) are provided by the European Centre for Medium-Range Weather Forecasts, from
their website at https:// www.ecmwf.int/en/forecasts/datasets/reanalysis-datasets/era5 (last ac- cess: 24 October 2022). Computer resources were provided by Consortium des Équipements de Calcul Intensif (CÉCI), funded by the Fonds de la Recherche Scientifique de Belgique (F.R.S. – FNRS) under grant no. 2.5020.11 and the Tier-1 supercomputer (Nic5) of the Fédération Wallonie Bruxelles infrastructure funded by the Walloon Region under grant agreement no. 1117545. Special thanks to Christoph Kittel for the help on the MAR data processing and evaluation.




**Code availability:** The code for the MAR data used in this study is tagged as v3.12 on https://gitlab.com/Mar-Group/MARv3 (MAR model, 2022). Instructions to download the MAR code are provided on https://www.mar.cnrs.fr (MAR Team, 2022).

**Data availability:** Elevation change fields are available via the World Data Center PANGAEA operated by AWI
Bremerhaven (after publication of the manuscript). The MAR outputs used in this study are available upon request by email (tdethinne@uliege.be).

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





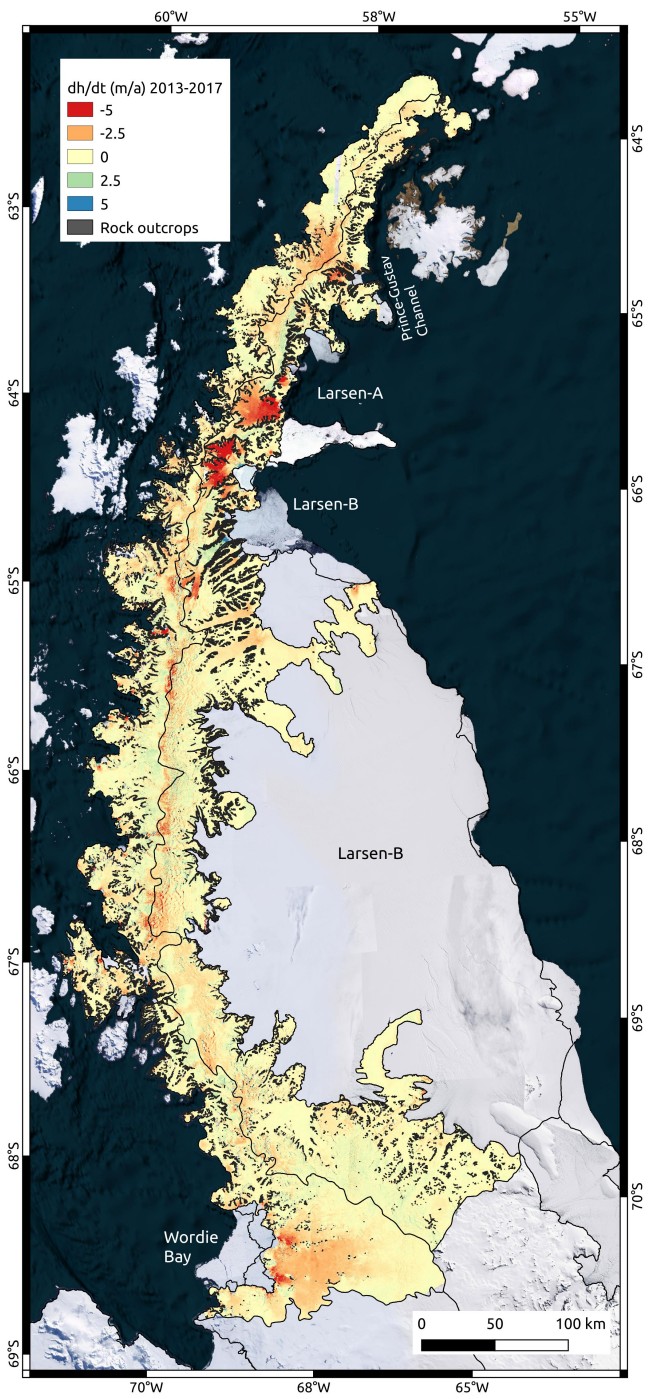

**Figure 1.** **Surface elevation changes between austral-winter of 2013 and 2017 derived from TanDEM-X acquisitions for the 25g**
**and 26g basin according to Zwally et al. (2012). Background: © Bing Satellite map by Microsoft; Black polygons: rock-outcrops**
**according to Silva et al. (2020)**





**Figure 2. a) Time difference between elevation measurements b) Mask of areas above 1800 m a.s.l. used for radar penetration bias correction. c) Election difference between refDEM and DEMs obtained in this study for 2013 and c) for 2017. Black polygons: rock-outcrops according to Silva et al. (2020)**



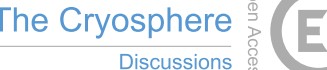

**Figure 3. Hypsometric distribution of measured (red bars) and total (grey bars) glacier area of 25g and 26g basins. Blue dots represent the mean Δh/Δt value in each elevation interval, including corrections in areas above 1800 m a.s.l. Black dots indicate the uncorrected Δh/Δt values. Grey areas mark the upper 1% quantile of the total glacier area distribution. Note: The scattered Δh/Δt values in areas above 2300 m a.s.l. are located around one peak towards the southern border of the study area, where the steep slope most likely lead to some biases. The affected area corresponds to only 0.06% of the total glacier area, and thus its impact on the total mass budget can be neglected**





**Figure 4. a) Average surface elevation changes (dh/dt) and b) total mass changes (dM/dt) for individual glaciers >20km²**
**Background: © Bing Satellite map by Microsoft**





**Figure 5. a) Total mass balance, b) mass balance ratio, c) imbalance ration, d) total climatic mass balance, e) total climatic mass balance anomalies, and f) specific mass balance for latitudinal subsets of the 25g and 26g drainage basins. Background: © Bing Satellite map by Microsoft; Black polygons: rock-outcrops according to Silva et al. (2020)**






**Figure 6. Sub-regions of the total study side based on a) Scambos et al. (2014), b) glacier front type, and c) Rott et al. (2018). Background: © Bing Satellite map by Microsoft; Black polygons: rock-outcrops according to Silva et al. (2020)**


**Figure 7. Grounding line positions at Hektoria-Green-Evans and Crane glaciers in 2016 from Rott et al. (2018), and b) Airy-Seller-Fleming Glacier for 2014 from Friedl et al. (2018) overlaid on the derived surface elevation changes between 2013 and 2017. Background: © Bing Satellite map by Microsoft; Black polygons: rock-outcrops according to Silva et al. (2020)**

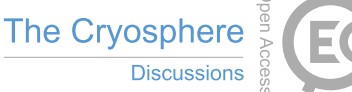

**Table 1: Summary of analyzed glacier area (S), glacier area covered by measurements (SM), average surface lowering rate (Δh/Δt) (note: the listed uncertainty of Δh/Δt represent the slope-weighted average offsets on rock-outcrops including consideration of spatial auto-correlation, but not SAR-signal penetration correction), total mass budget (ΔM/Δt), average climatic mass balance (CMB), imbalance ratio (IR), and mass balance ratio (MBR) for different basin and subregion definitions. \*Subregion definitions according to Scambos et al. (2014), +Subregion definitions based on glacier front type: No IS: non ice shelf tributaries, IS break:**
**former ice shelf tributaries, IS no break: current ice shelf tributaries; °Subregion definition according to Rott et al. (2018)**

| Basin/Subreg. | Lat. | $S$ (km²) | $S_m$ (km²) | $\Delta h/\Delta t$ (m/a) | $\Delta M/\Delta t$ (Gt/a) | CMB (m w.e./a) | CMB (Gt/a) | IR | MBR |
|---|---|---|---|---|---|---|---|---|---|
| 25g&26g | total | 74888.0 | 72222.1 | -0.283±0.003 | -24.1±2.8 | 1.72 | 128.0 | -0.19 | 0.19 |
| | total | 34474.5 | 32952.6 | -0.295±0.007 | -12.2±1.7 | 2.21 | 71.9 | -0.17 | 0.43 |
| | 63 | 3184.2 | 3035.2 | -0.341±0.023 | -1.0±0.1 | 1.80 | 5.4 | -0.18 | -0.34 |
| | 64 | 4273.1 | 4068.8 | -0.354±0.023 | -1.4±0.2 | 2.39 | 10.1 | -0.14 | -0.20 |
| 25g | 65 | 4903.3 | 4545.5 | -0.253±0.028 | -1.1±0.4 | 2.74 | 13.4 | -0.08 | 0.46 |
| | 66 | 5607.8 | 5404.5 | 0.079±0.023 | 0.4±0.4 | 2.92 | 16.1 | 0.02 | -3.03 |
| | 67 | 4195.1 | 3854.6 | -0.283±0.018 | -1.1±0.1 | 1.61 | 7.1 | -0.15 | 0.73 |
| | 68 | 3511.7 | 3430.0 | -0.042±0.017 | -0.1±0.1 | 1.58 | 5.6 | -0.02 | 7.75 |
| | 69 | 8799.2 | 8613.9 | -0.607±0.010 | -4.8±0.6 | 1.61 | 14.1 | -0.34 | 0.49 |
| | total | 40413.5 | 39269.5 | -0.276±0.004 | -12.0±1.2 | 1.34 | 57.4 | -0.21 | -0.04 |
| | 63 | 2185.8 | 2139.0 | -0.261±0.008 | -0.5±0.0 | 1.68 | 3.6 | -0.14 | -0.92 |
| | 64 | 6030.0 | 5674.3 | -0.950±0.012 | -5.2±0.4 | 1.89 | 11.5 | -0.45 | -0.21 |
| 26g | 65 | 5467.8 | 5308.6 | -0.390±0.016 | -1.9±0.2 | 1.41 | 8.1 | -0.24 | -0.19 |
| | 66 | 8578.0 | 8418.7 | -0.101±0.009 | -0.8±0.3 | 1.29 | 11.1 | -0.07 | 0.17 |
| | 67 | 4174.9 | 3950.5 | -0.364±0.018 | -1.4±0.1 | 2.00 | 8.2 | -0.17 | 0.45 |
| | 68 | 7581.2 | 7411.0 | -0.164±0.009 | -1.1±0.1 | 1.33 | 10.1 | -0.11 | 0.47 |
| | 69 | 6395.8 | 6367.3 | 0.137±0.003 | 0.8±0.3 | 0.72 | 4.6 | 0.17 | -0.20 |
| I-Ipp | total | 58985.4 | 56351.5 | -0.277±0.004 | -14.7±2.0 | 1.83 | 108.3 | -0.14 | 0.13 |
| | 63 | 5371.9 | 5173.3 | -0.309±0.006 | -1.5±0.1 | 1.72 | 9.0 | -0.17 | -0.54 |
| | 64 | 10537.8 | 9798.7 | -0.690±0.010 | -6.6±0.6 | 2.04 | 21.5 | -0.30 | -0.21 |
| | 65 | 10272.8 | 9779.8 | -0.344±0.012 | -3.2±0.5 | 2.02 | 21.3 | -0.15 | 0.05 |
| | 66 | 14570.6 | 14180.4 | -0.028±0.007 | -0.4±0.7 | 1.88 | 27.1 | -0.01 | 3.36 |
| | 67 | 8665.2 | 8067.3 | -0.320±0.012 | -2.5±0.2 | 1.76 | 15.5 | -0.16 | 0.55 |





| | | | | | | | | |
|---|---|---|---|---|---|---|---|---|
| 68 | 9567.2 | 9352.0 | -0.073±0.010 | -0.6±0.1 | 1.46 | 14.0 | -0.05 | 2.19 |
| nAP* | 26325.3 | 25191.1 | -0.463±0.006 | -11.0±1.2 | | | | |
| nAP East* | 13371.2 | 12921.7 | -0.651±0.008 | -7.8±0.6 | | | | |
| nAP North* | 1310.8 | 1302.5 | -0.067±0.004 | -0.1±0.0 | | | | |
| nAP West* | 11643.3 | 10966.8 | -0.286±0.016 | -3.0±0.7 | | | | |
| No IS[+] | 28445.0 | 27149.4 | -0.171±0.005 | -4.4±1.2 | | | | |
| IS break[+] | 58628.1 | 51064.4 | -0.316±0.003 | -16.7±2.3 | | | | |
| IS no break[+] | 34541.7 | 30460.6 | -0.123±0.003 | -3.8±1.1 | | | | |
| Rott18° | 11778.3 | 11347.7 | -0.692±0.009 | -7.3±0.6 | | | | |



**Table 2: Comparison of glacier mass balances obtained in this study and by Rignot et al. (2019) for the period 2013-2017. S: glacier area; ΔM/Δt: total and specific mass balance. *assumed balanced mass budget by Rignot et al. (2019), +excluding glaciers with assumed balanced mass budget**


| Glacier | This study | | | Rignot et al. (2019) | | | difference | |
|---|---|---|---|---|---|---|---|---|
| | $S$ (km²) | $\Delta M/\Delta t$ (Gt/a) | $\Delta M/\Delta t$ (kg/m²/a) | $S$ (km²) | $\Delta M/\Delta t$ (Gt/a) | $\Delta M/\Delta t$ (kg/m²/a) | $S$ (%) | $\Delta M/\Delta t$ (%) |
| Crane | 1139 | -0.57±0.06 | -0.50 | 1216 | -1.11±0.18 | -0.91 | -6.7 | 83.4 |
| Drygalski | 945 | -1.96±0.14 | -2.07 | 996 | -4.29±0.80 | -4.31 | -5.4 | 107.8 |
| Flask | 1137 | -0.34±0.04 | -0.30 | 1178 | -0.53±0.18 | -0.45 | -3.6 | 50.8 |
| Fleming | 7710 | -4.92±0.55 | -0.64 | 8988 | -8.02±2.77 | -0.89 | -16.6 | -39.8 |
| HGE | 1401 | -2.51±0.19 | -1.79 | 1431 | -7.24±0.31 | -5.06 | -2.1 | 183.0 |
| Jorum | 417 | -0.13±0.03 | -0.31 | 520 | -0.21±0.06 | -0.40 | -24.6 | 28.1 |
| Leppard | 1566 | -0.55±0.06 | -0.35 | 1812 | -0.30±0.22 | -0.16 | -15.7 | -53.6 |
| Mapple* | 130 | -0.03±0.01 | -0.26 | 187 | 0.00±0.07 | 0.00 | 0.0 | -100 |
| Melville | 187 | -0.09±0.01 | -0.49 | 281 | -0.10±0.00 | -0.36 | -50.1 | -28 |
| Pequod* | 229 | -0.01±0.02 | -0.06 | 327 | 0.00±0.01 | 0.00 | -43.3 | -100 |
| Punchbowl* | 108 | -0.02±0.01 | -0.18 | 99 | 0.00±0.01 | 0.00 | 8.3 | -100 |
| Starbuck* | 253 | 0.01±0.01 | 0.03 | 280 | 0.00±0.02 | 0.00 | -10.9 | -100 |
| Stubb* | 177 | -0.01±0.02 | -0.07 | 196 | 0.00±0.03 | 0.00 | -10.9 | -100 |
| Total | 15399 | -11.14±1.16 | -0.72 | 17512 | -21.80+ -4.66 | -1.25 | -13.7 | 72% |
| Total+ | 14503 | -11.07±1.08 | -0.76 | 16421 | -21.80+ -4.53 | -1.32 | -13.2 | 74% |