# Peer review of "Mass changes of the northern Antarctic Peninsula Ice Sheet derived from repeat bi-static SAR acquisitions for the period 2013-2017"

_The Cryosphere, 2022_

## Author Comment (AC1)

**Answers to Reviewer 1:**

**First of all we want to thank the reviewer for the general positive feed back and constructive comments on our manuscript. All comments have been taken into account and a list of answers and undertaken actions is given below. Answers are marked in "blue".**

**Comments by Reviewer 1:**

The introduction looks good. It provides a comprehensive overview of the current state of the Antarctic Peninsula (AP) ice sheet, including the effects of climate change on temperature, ice shelf loss, and ice mass loss. The introduction also highlights the limitations of current methods used to estimate ice mass loss on the AP and identifies the need for further research, such as geodetic mass balance estimation. Overall, the introduction provides a clear and concise background to the study area and sets the stage for the research that follows.

The data used for the study appears to be scientifically sound and coherent. The authors used bistatic Synthetic Aperture Radar (SAR). They used two coverages of the AP with TDX data acquisitions from austral-winter 2013 and 2017 for their analysis, and a reference DEM (refDEM) based on the global TanDEM-X DEM at 12 m spatial resolution.

To obtain information on the CMB, the authors used output from the regional climate model MAR.

The authors computed the average CMB for the period July 2013 until June 2017 and the absolute and relative difference (dCMB) in respect to whole temporal coverage of the MAR data (2008-2022) is computed to obtain information CMB anomalies during the study period. They also defined the mass balance ratio (MBR) by dividing the CMB anomalies (dCMB) by the total mass change ($\Delta M/\Delta t$) to indicate the contribution of CMB changes on the mass change.

Overall, the data used in the study appears to be based on established and widely accepted scientific methods and models, and the authors have taken care to provide appropriate references and explanations for their methods.

The method seems comprehensive and well thought out. The differential interferometric SAR processing approach used to derive DEMs from TDX data is well explained, along with the advantages of this approach. The iterative coregistration procedure used to generate smooth DEM mosaics for each time step is also explained in detail. The text describes how the coregistration procedure had to be adapted for the study area due to the limited availability of ice-free areas and complex topography. The methods used to compute ice mass change rates are also described, including how different basin definitions and sub-region definitions were used, and how voids in the elevation change field were filled. The text explains how the results were converted to ice mass changes using a volume-to-mass conversion factor. The method seems well developed and comprehensive.

The article discusses the variation in ice surface elevation in the northern part of Antarctica and its possible causes. The paper presents an analysis of changes in the surface elevation of glaciers and ice basins over an area of 770,000 km², covering 96.4% of the glaciated area of northern Antarctica. The results indicate a significant loss of ice mass, with loss rates of up to -8 m/yr on some glaciers and a total mass loss of -24.3 ± 5.8 Gt/yr in the study region. Possible causes for this ice mass loss are discussed based on previous studies, including warming ocean waters and changes in snow

accumulation. However, the study highlights that additional analyses of ice dynamics are needed to confirm the conclusions drawn from variations in ice surface elevation. A limitation of the study is that it focuses only on the selected study region and does not consider the long-term temporal variation in Antarctic ice changes.

We understand the reviewer's concern. However, the data handling and the processing resources needed for carrying out such an analysis on Antarctic wide scales would be too difficult, at least using our computing infrastructure. Moreover, for most regions in Antarctica the amount of ice free areas for coregistration is very low, which would further strongly hamper the analysis. However, there exists estimates based on other approaches, like altimetry or gravimetry, for the remaining regions in Antarctica. These approaches have certain limitations on the AP, but work very well at the less complex topography of e.g. the East or West Antarctic Ice Sheet (see e.g. the IMBIE reports).

One potential critique is that the study relies solely on bi-static SAR data, which may have limitations in terms of accuracy and resolution compared to other measurement methods.

As also requested by Reviewer 2, we evaluated the quality and suitability of the TanDEM-X data for assessing the volume changes using independent data sources. For more details see the (new) supplement and the answers to Reviewer 2.

The authors acknowledge that there is a need for improved ice thickness data towards the grounding line, which is a dominant error source in ice discharge estimates on the AP.

Additionally, while the study provides a detailed assessment of glacier mass balances at unprecedented spatially detailed scales and with high precision, there may be other factors that are not accounted for in the analysis. For example, the study identifies that most of the mass losses are caused by ice dynamic changes, but it may be difficult to distinguish between changes in ice dynamics caused by climate variability versus other factors such as ocean currents or internal ice sheet processes.

We agree with the reviewer, and that is why we also pointed out in the manuscript that additional research on the glacier ice flow evolution (seasonal and long-term) is needed to better assess the driving factors.

The study provides important new insights into glacier mass balance in the northern AP and is significant for cryosphere science, I advise that it be accepted for publication.

---

## Author Comment (AC2)

**Answers to Reviewer 2:**

**First of all we want to thank the reviewer for constructive comments on our manuscript. All comments have been taken into account and a list of answers and undertaken actions is given below. Answers are marked in "blue"**

**NOTE: For better clarification of the answers, we added the new figures from the supplement at the end of this document, since we need to upload the revised manuscript and the supplement at a later stage.**

Comments by Reviewer 2:

This study describes the derivation of elevation and mass changes for the entire northern Antarctic Peninsula between 2013 and 2017 based on bistatic interferometric SAR data from the TanDEM-X mission. Using a regional climate model, information on climate mass balance have been related to derived geodetic mass balance estimates in order to identify potential drivers of melt.

In general, this study is of interest, since information on elevation and mass change for the Antarctic Peninsula at such high spatial resolution could contribute to a better understanding of the interactions between climate warming and the dynamics of the Antarctic Ice Sheet.

However, the manuscript raises many questions about the data and methodology as well as considerable concerns about the validity of the results. One of the main problems of this study is the lack of quantitative validation of the derived elevation data. A detailed interpretation of the results, as well as the linkage with climate models (as in Section 4) only appears meaningful if the validity of the previously derived results on elevation change have been verified. Therefore, I cannot recommend this manuscript in its present form for publication in The Cryosphere. I suggest that the authors seriously address the following issues before re-submission:

1. In general, the input data and the methodology are described insufficiently in many parts of Section 2 and Section 3. There is no detailed information provided regarding the acquisition parameters (e.g. acquisition time, incidence angle, orbit, beam) of the TanDEM-X data. However, this information is of great interest for assessing the accuracy of the elevations represented by the mosaics. It would also be very helpful to provide an overview of the TanDEM-X footprints in Figure 2.

According to the reviewer comment, we added a table of all used TanDEM-X data as supplement. The table contains information regarding acquisition time, incidence angle, orbit, beam/strip as well as interferometric parameters like effective baseline and height of ambiguity. Moreover we added the respective TanDEM-X foot prints in Figure 2c) and d) as requested also further below. Regarding the adjustments in Section 2 and 3 see below.

2. The description of the DEM generation (p.4, l120) is insufficient and far too short and the individual processing steps are not comprehensible for the reader. I suggest to include in a more detailed description of the interferometric SAR processing, possibly in combination with a flowchart.

We followed the reviewers suggestions and added a more detailed description of the interferometric SAR processing, even-though it does not differ from the linked literature (Sommer et al. 2022) and citations there in.
The following section was added:

"… The phase-to-height sensitivity is computed by means of a simulated differential interferogram derived from the refDEM and the refDEM lowered by 100 m. Subsequently the unwrapped differential interferogram based on the TDX is data is converted to a differential elevation map by applying the derived phase-to-height sensitivity information. Finally, the height information of the refDEM is added to compute absolute heights and the resulting DEM is orthorectified and geocoded….."

3. Generation of such large mosaics covering the entire Antarctic Peninsula faces numerous challenges, as the authors confirm (lack of ice-free stable terrain for co-registration, mosaicking of multiple coverages in overlapping areas, treatment of height offsets between the individual DEMs, etc.). In this regard, it is indispensable to establish the accuracy of the derived elevation data through quantitative validation using independent height measurements and to address the horizontal and vertical accuracy of the whole mosaic and its limitations. However, a quantitative validation is completely missing. Consequently, the reliability of the results is in question. For validation purposes, time stamped REMA strips or IceBridge data might be a useful source of reference heights.

We understand the reviewers concerns and included an evalutation of the derived TanDEM-X DEM heights using independent measurements. Since the Tandem-X data used withing this study was acquired during austral-winter there exists no contemporaneous IceBridge and only very limited contemporaneous time stamped REMA data.
A detailed description of the evaluation and additional figures are added as supplement to the manuscript, with respective cross-links and statements in the main manuscript. A brief description is added in the following:

Comparison to IceBridge IDHDT4 data (similar to Rott et al. (2018):

IceBridge IDHDT4 surface elevation change rate data from a longer and shifted observation period (2011-2016) was employed. The IDHDT4 data covers only limited sections of our study area (Fig. S1). Similar to Rott et al. (2018), longitudinal profiles of outlet glaciers were selected, and very crevassed or steep slope areas and regions affected by front position changes were masked out.
In contrast to Rott et al. (2018) our observation period did not fit the IceBridge IDHDT4 period. Consequently, considerable offsets between TanDEM-X and IDHDT4 dh/dt measurements are revealed at some glaciers with dynamic change rates Consequently, these glaciers were not considered in the comparison of elevation change rates from TanDEM-X (2013-2017) and IDHDT4 (2011-2016). In total 1592 samples were evaluated. In Figure S1 the spatial distribution and the elevation change rate offsets between both datasets is illustrated. The scatter plot in Figure S2 illustrates the concordance of the dh/dt measurements. A mean offset of 0.12 m/a and a RMSE of 0.337 m/a is obtained. Both values are higher than the findings revealed by Rott et al. (2018) (mean offset of -0.08 m/a and RMSE of 0.20 m/a). However, in contrast to Rott et al. (2018), there is a

temporal difference between both elevation change rate data sets for this study, most likely explaining the more pronounced differences as compared to Rott et al. (2018).

Comparison to time stamped REMA DEMs:

Contemporaneous time stamped REMA tiles for both TanDEM-X acquisition periods were downloaded a 2 m spatial resolution and bi-linearly resampled to the resolution of the TanDEM-X data (30 m). A temporal threshold of maximum 30 days between the individual TanDEM-X acquisitions and the REMA data was applied. Subsequently, the provided REMA bitmask were applied to remove unreliable or interpolated information. Subsequently, the REMA DEM tiles were coregistered following the approach used for the TanDEM-X raw DEMs. Finally, the offset between the REMA and TanDEM-X surface elevation data sets were computed.
The REMA DEM coverages for both periods (2013 and 2017) do not overlap. Consequently, we analyzed each period independently. In Figure S4 the hypsometric distribution of the offsets are illustrated. Negative values indicate that TanDEM-X heights were below the REMA surface, which serves as an indicator for SAR signal penetration. The offsets between REMA and TanDEM-X in 2013 show no trend with elevation, where as for 2017 an slight trend towards higher offsets in regions above ~1800 m a.s.l. is visible. This finding supports our assumption of a potential SAR signal penetration offset between both acquisitions for elevated regions. (Note: The offsets are measured at different locations and thus SAR signal penetration offset between our TanDEM-X coverage in 2013 and 2017 can not be derived from this data, since the SAR signal penetration can vary strongly on spatial scales, e.g. due to different accumulation rates or melt rates).
However, we concluded, that the TanDEM-X and REMA surface elevations agree quite well (<1 m average offset, which we attribute to SAR signal penetration) for both periods, demonstrating the suitability of our TanDEM-X DEMs as elevation information source for the study region.

4. Another important point that receives far too little attention in the manuscript is the elevation bias due to signal penetration, which can have enormous effects on the accuracy of the elevation and mass changes. There are several studies demonstrating a significant bias in TanDEM-X elevation data due to signal penetration (Rizzoli et al., 2017; Abdullahi et al., 2019; Fischer et al., 2019, 2020; Rott et al., 2021; Wessel et al., 2021). Some studies suggest substantial seasonal variation in signal penetration from elevation, but also from year to year (e.g., from winter to winter, as in the present study). The assumption of a linear increase of the penetration bias above a certain altitude presumably cannot adequately reflect the complex interaction between (sub-)surface structures and continuously changing acquisition geometry. In this context, the validity of penetration correction model has to be demonstrated using independent reference data.

As mentioned in the previous comment. There exists only limited contemporaneous, interdependent height data, which fits to the used TanDEM-X acquisitions from austral winters 2013 and 2017. The SAR signal penetration can vary on regional scales depending on different climatic settings (e.g. local accumulation rates). Unfortunately, the independent height data covers different regions in 2013 and 2017, hampering a meaningful assessment of the SAR signal penetration differences between both TanDEM-X acquisition periods. Consequently, the assessment or validation of our applied correction for the SAR signal penetration for our used TanDEM-X acquisitions is not possible using other height sources, and we applied the approach also used in previous analyses.

5. In addition, the authors have made some assumptions without demonstrating their validity:

- **p.3, line 84f**: The authors argue negligible signal penetration when comparing winter season data and refer to a study from Rott et al. (2018). However, this study points out that the negligibility of signal penetration is due to the same acquisition geometry (similar incidence angle, orbit and beam) of the TanDEM-X data, which is supported by the similarity of the backscatter coefficients of the individual scenes. It would be a possibility to analyze the backscatter coefficients of the used data to make a statement about the similarity of the backscatter behavior and thus the signal penetration.

As suggested by the reviewer, we carried out an analysis of the SAR backscatter for both acquisition periods. We carried out a similar analysis as done by Sommer et al. (2022) and plotted the distribution on the SAR backscatter vs. elevation (see supplement). In Figure S5 the hypsometric distributions of the mean SAR backscatter values are illustrated for both acquisition periods. Both plots show a similar distribution with an increased offset towards higher elevations (above ~1800 m a.s.l.), supporting our assumption of a potential SAR signal penetration bias at higher elevations. Thus, we conclude that the assumption of negligible signal penetration differences for acquisitions from winter season, following Rott et al. (2018), is valid for the lower sections of our study site, where we did not apply any correction.

- **p.5, lines 145-148**: Considering Figures 2a, c, and d, structures can certainly be guessed at, which are due to elevation differences caused by the different penetration biases in the individual DEMs. Integrating the footprints of the individual DEMs in Figure 2 would be important to support the authors' statement, rather than just saying that 'the pattern does not match the outline of the individual DEMs'.

As requested by the reviewer, we added the footprints of the individual DEMs in Figure 2c and d.

- **p.6, lines 165ff**: How do the authors conclude that signal penetration increases linearly between 1800 and 2400 m a.s.l.? What about the areas below and above this altitude level? Are these assumptions based on experience, analysis of data, other studies based on interferometric X-band SAR data conducted on the Antarctic Peninsula?

We applied a increasing correction value following previous studies on using InSAR data for geodetic mass balance measurements. We are aware, that the processes leading to differences in SAR signal penetration do not follow linear trend towards higher elevations. However, the major factor regarding SAR signal penetration is the amount of liquid water in the firn pack, which typically decreases towards higher altitudes. Consequently, we used this approach even-though it is just a rough approximation.

The upper limit was selected based on the hypsometric distribution of the ice elevations, see Figure 3. Only two small peaks stretch above this limit. Consequently, they were neglected for the definition of the upper limit. The lower bound was defined by the inspection of the elevation change patterns, see Sections 3, the comparison with REMA data and the SAR backscatter analysis (see above). Regions which are potentially affected by SAR signal penetration were identified above this

elevation. Moreover, the elevation change distribution in Figure 3 shows a  clear break in slope at 1800 m. We adjusted the wording in Section 3 to be more precise.

"….Even though, solely TDX austral winter data is used in this analysis, some elevation change patterns in upper glacier regions (see Figure 1 and break in slope of the elevation change data in Figure 3, as well as the comparison to independent elevation data (Supplement section 1.2) and the analysis of the SAR backscatter signal (Supplement section 2)) seems to be caused by differences in the SAR signal penetration between the acquisitions. These areas are located at elevations above 1800 m a.s.l. covering an area of 12.306 km², corresponding to 16.4% of 25g & 26g drainage basins. In order to correct for these potential offsets, we applied a linear increasing correction of dh for the elevation range from 1800 – 2400 m a.s.l. of up to 2 m, similar to Braun et al. (2019) or Seehaus et al. (2020a), leading to correction of the volume change by -2.76 km³ and an average elevation change by 0.04 m/a (25g & 26g basins). The upper limit of 2400 m a.s.l. was chosen since only two small peaks in the southern part of the study area stretch above this limit. …"

**References**

Abdullahi, S., Wessel, B., Huber, M., Wendleder, A., Roth, A., Kuenzer, C., 2019. Estimating Penetration-Related X-Band InSAR Elevation Bias: A Study over the Greenland Ice Sheet. Remote Sens., Remote Sensing 11, 2903.

Fischer, G., Jäger, M., Papathanassiou, K.P., Hajnsek, I., 2019. Modeling the Vertical Backscattering Distribution in the Percolation Zone of the Greenland Ice Sheet With SAR Tomography. IEEE J. Sel. Top. Appl. Earth Obs. Remote Sens., IEEE Journal of Selected Topics in Applied Earth Observations and Remote Sensing 12, 4389–4405.

Fischer, G., Papathanassiou, K.P., Hajnsek, I., 2020. Modeling and Compensation of the Penetration Bias in InSAR DEMs of Ice Sheets at Different Frequencies. IEEE J. Sel. Top. Appl. Earth Obs. Remote Sens., IEEE Journal of Selected Topics in Applied Earth Observations and Remote Sensing 13, 2698–2707.

Rizzoli, P., Martone, M., Rott, H., Moreira, A., 2017. Characterization of Snow Facies on the Greenland Ice Sheet Observed by TanDEM-X Interferometric SAR Data. Remote Sens., Remote Sensing 9, 315.

Rott, H., Scheiblauer, S., Wuite, J., Krieger, L., Floricioiu, D., Rizzoli, P., Libert, L., Nagler, T., 2021. Penetration of interferometric radar signals in Antarctic snow. The Cryosphere 15, 4399–4419.

Sommer, C., Seehaus, T., Glazovsky, A., Braun, M.H., 2022. Brief communication: Increased glacier mass loss in the Russian High Arctic (2010–2017). The Cryosphere 16, 35–42. https://doi.org/10.5194/tc-16-35-2022

Wessel, B., Huber, M., Wohlfart, C., Bertram, A., Osterkamp, N., Marschalk, U., Gruber, A., Reuß, F., Abdullahi, S., Georg, I., Roth, A., 2021. TanDEM-X PolarDEM 90 m of Antarctica: generation and error characterization. The Cryosphere 15, 5241–5260.

[Figure]

**Figure S1: Spatial distribution of the offset between the TanDEM-X elevation change rates and the IceBridge ATM IDHDT4 data. Background: © Bing Satellite map by Microsoft**

[Figure]

**Figure S2: Scatter plot of TanDEM-X (TDX) and IceBridge ATM IDHDT4 elevation change measurements.**

[Figure]

**Figure S3: Spatial coverage of elevation differences between contemporaneous TanDEM-X and REMA DEMs for 2013 (blue polygon) and 2017 (red polygon).**

[Figure]

**Figure S4: Hypsometric distribution of differences between contemporaneous TanDEM-X and REMA DEMs for 2013 (blue) and 2017 (green). Whiskers indicate root mean square differences (RMSD) for the respective elevation bin**

[Figure]

**Figure S5: Hypsometric distribution of SAR backscatter values for 2013 (blue) and 2017 (orange). Whiskers indicate normalized mean absolute deviation (NMAD). The gray area marks the upper 1% quantile of the total glacier area distribution (see Figure 3 in the main manuscript)**

---

## Author Response (AR2)

**Answers to Reviewer 1 (second iteration):**

**First of all, we want to thank the reviewer for the positive feedback and constructive comments on our manuscript. All comments have been taken into account and a list of answers and undertaken actions is given below. Answers are marked in "blue".**

**Comments by Reviewer 1:**

I would like to thank the authors for addressing all of my comments, especially the quantitative validation of the derived TanDEM-X DEM mosaics.
I recommend the manuscript for publication pending some points below, as well as a final proofread on typos which I leave up to the authors.

Sorry for the issues. A spell and grammar check was carried out before resubmission.

Main manuscript
L 179 ff Please quantify the agreement of the TanDEM-X DEM mosaics with the independent reference data. Consider providing mean and standard deviation of the residuals between TanDEM-X DEM mosaics and IceBridge and REMA elevations.

According to the reviewer's suggestion, we added the respective values: "TanDEM-X to Operation IceBridge: mean offset of 0.12 m/a, RMSD of 0.34 m/a; TanDEM-X to REMA for 2013 and 2017: mean offset -0.47 m/a and RMSD of 3.11 m/a"

L 183 ff From my point of view, neither Figure S4 nor Figure S5 clearly support the assumption that the penetration bias increases linearly between 1800 and 2400 m a.s.l.. Please clearly point out that this assumption is a rough simplification, with reference to the current literature.

We agree with the reviewer and changed the wording to: "In order to correct for these potential offsets, we applied a simple correction approach, assuming a linear increasing penetration correction of dh for the elevation range from 1800 – 2400 m a.s.l. of up to 2 m, similar…."

Supplement
1. Please discuss the spread in Figure S2.

On page 2 of the supplement, the difference between the TanDEM-X and IceBridge data, leading to the spread in Figure S2 is described:
"In particular the offsets in *dh/dt* in the upper reaches of Leppard and Attlee Glacier lead to the strongly negative offsets (see Figure S1 and S2) and increased RMSD value. These areas of the AP plateau receive higher accumulation rates than the low-lying areas on the east coast (e.g. van Wessem et al., 2015). Subsequently, surface elevations can vary strongly on inter-annual but also sub-annual scales, depending on the accumulation rates and timing of data acquisition. On the other hand differences in the SAR signal penetration between the TanDEM-X acquisitions in 2013 and 2017 can also lead to such offsets.."

2. In Figure S4, the elevation differences between the TanDEM-X DEM mosaics and the REMA DEM show a slight increase at higher elevations for 2017, but not for 2013. Thus, this graph does not sufficiently support the assumption of a linearly increasing penetration bias at higher elevations for the study area.

The reviewer is right. However, we did not use Figure S4 to justify the penetration bias correction. As stated in the supplement, it just "…. supports our assumption of a potential SAR signal penetration offset between both acquisitions for elevated regions. (Note: The offsets are measured at

different locations and thus SAR signal penetration offset between our TanDEM-X coverage in 2013 and 2017 can not be derived from this data, since the SAR signal penetration can vary strongly on spatial scales, e.g. due to different accumulation rates or melt rates).… "

Moreover, we adjusted the wording in the main manuscript, to more clearly state that our correction is a simplification and just an approach (see answer to comment above).